# Public Perceptions of Contentious Risk: The Case of Rubber Granulate in the Netherlands

**DOI:** 10.3390/ijerph16122250

**Published:** 2019-06-25

**Authors:** Marion de Vries, Liesbeth Claassen, Marcel Mennen, Aura Timen, Margreet J. M. te Wierik, Danielle R. M. Timmermans

**Affiliations:** 1Centre of Environmental Safety and Security, National Institute for Public Health and the Environment (RIVM), 3720 BA Bilthoven, The Netherlands; Liesbeth.claassen@rivm.nl (L.C.); marcel.mennen@rivm.nl (M.M.); 2Centre for Infectious Disease Control, National Institute for Public Health and the Environment (RIVM), 3720 BA Bilthoven, The Netherlands; aura.timen@rivm.nl or a.timen@vu.nl (A.T.); margreet.te.wierik@rivm.nl (M.J.M.t.W.); 3Department of Public and Occupational Health, Amsterdam UMC, Vrije Universiteit Amsterdam, Amsterdam Public Health research institute, 1007 MB Amsterdam, The Netherlands; drm.timmermans@vumc.nl; 4Athena Institute for Innovative and Transdisciplinary Research in Health Sciences VU University Amsterdam, 1081 HV Amsterdam, The Netherlands; 5National Institute for Public Health and the Environment (RIVM), 3720 BA Bilthoven, The Netherlands

**Keywords:** risk perception, risk communication, environmental health risk, rubber granulate, crumb rubber

## Abstract

This paper reports on the perceptions of risk related to practicing sports on fields containing rubber granulate infill, and preferences for mitigation measures, among people with and without offspring exposed to rubber granulate. Two repeated surveys were conducted among members of the general population and parents of children aged under 18, in the middle of a dynamic public discussion about the potential health risks of exposure to rubber granulate. The first survey (N = 1033) was administered in December 2016 at a time characterized by considerable public uncertainty and contrasting opinions in the public risk debate. The second survey (N = 782) was conducted in January 2017 after the publication of a risk assessment report, which concluded that practicing sport on fields containing rubber granulate is safe. Multilevel analyses were performed to study changes in perceptions of risk and mitigation preferences in the time between the two surveys, the influence of being familiar with new information following the risk assessment report, and the differences in the perceptions of risk and mitigation preferences between groups with and without offspring exposed to rubber granulate. The results of this study show that, initially, a substantial proportion of the Dutch public perceived practicing sports on fields containing rubber granulate as a potential health threat to children. Over time, after publication of a new risk assessment study stating that practicing sports on fields containing rubber granulate is safe, perceived risk and preferences for mitigation of this risk decreased, especially among those who were familiar with the new information. Parents of children under the age of 18, in particular those with children who were exposed to rubber granulate, were more likely to perceive the risk as higher and to prefer a stricter mitigation policy. These insights may be important to inform public health communication strategies with respect to the timing and tailoring of risk messages to various groups.

## 1. Introduction

### 1.1. The Case of Rubber Granulate in the Netherlands

In October 2016, questions were raised in the Netherlands about the potential health risks of practicing sports on fields containing rubber granulate infill. In the Netherlands, a large proportion of the sport fields have rubber granulate infill and many children play soccer on these fields (see Box 1 for a brief overview of the origin and use of rubber granulate). The public interest in this risk topic followed an episode of a weekly television programme [1], which raised the suggestion that rubber granulate would be the cause of children developing cancer. In the following weeks, headlines such as “Carcinogenic substances at almost all soccer fields” [2] (and “’Playing soccer on rubber granulate is experimenting with healthy people’” [3] were all over the news.

### 1.2. Concerns About Contemporary Risks from the Environment, and How We Deal with Them

The potential health risk posed by rubber granulate is one of many contemporary man-made risks from the environment that trigger public debate. Comparable public debates arose around, for example biotechnology [4], electromagnetic fields [5] and food additives [6]. Public debates about these so called ‘modern risks’ are often characterised by concerns about invisible hazardous substances, and potential, though often highly uncertain, health consequences [7,8]. These public concerns about risks can have considerable societal impact. Public concerns have shown to be, although not univocal, associated with individual risk (mitigation) behaviour and support for risk management [9,10].

It is common for authorities to address public concerns about risks by initiating more research on the particular risk. In the case of rubber granulate, several literature reviews and additional risk assessments by public health institutes in the Netherlands, and in other countries, were (and are currently being) conducted to provide clarity about the risks [11,12,13,14,15,16]. The overall conclusion of these studies is that there are no serious health risks to be expected as a result of practicing sports on fields containing rubber granulate. Nevertheless, conclusions like this following a literature review or risk assessment always bring with them a certain level of uncertainty. This uncertainty is either because of study limitations (e.g., a limited sample size or a lack of longitudinal data) or simply because of scientific nuance (e.g., a common scientific phrase is ‘a risk cannot be excluded’) [17,18,19].

Whilst uncertainty lies at the core of science, lay people generally feel uncomfortable with uncertainties. When people perceive uncertainties around risks, this discomfort can lead to higher levels of perceived risks [20] and increased demands for risk mitigation [21]. Scientific risk assessments alone will therefore probably not adequately meet the public’s need for information and reassurance, and might even increase concerns and the demand for risk mitigation because of their inherent uncertainties. To address the public’s information needs and concerns, we need to carefully develop risk communication that can bridge the gap between scientific assessments of risk and the public’s beliefs about these risks [22].

Box 1Rubber granulate: a brief overview.Rubber granulate (or crumb rubber) is a material used as 
infill for synthetic turf fields. These fields are an alternative for sport 
fields containing natural grass. Synthetic turf fields are popular as they are 
more easily maintained than fields containing natural grass, and can be played 
on in various weather circumstances. Rubber granulate is used as infill of 
~1800 synthetic turf fields in the Netherlands, which are used for practicing 
sports; mainly soccer. Rubber granulate makes synthetic turf fields more 
similar in use to fields containing natural grass.Most rubber granulate infill is made out of recycled car 
tyres. This type of rubber granulate is called styrene butadiene rubber 
granulate (SBR). Car tyres contain various chemical, and potentially hazardous, 
substances, such as aromatric hydrocarbons (PAHs), various heavy metals, 
plasticisers and (semi-)volatile compounds [23]. Some of these substances are added to the 
rubber to create well-functioning tyres, and some are the result of the 
manufacturing process.Because of the presence of these substances in rubber 
granulate, various studies have been performed to investigate whether there are 
health risks related to practicing sports on fields containing rubber granulate 
infill. A review of these studies concluded that overall there is no elevated 
health risk to be expected [24]. Nevertheless, some of the studies included in 
the review, pointed out a number of uncertainties because of study limitations 
(e.g. limited exposure data, sample size, and substances investigated). Because 
of these uncertainties, additional research on the safety of use of rubber 
granulate infill was recommended.

### 1.3. Understanding Perceptions of Risk

In order to develop effective risk communication, and support the public to make well-informed decisions about risks, an understanding of perceptions of risks is essential. Perceptions, including perceptions of risk, are “mental processes through which a person takes in, deals with and assesses information from the environment (physical and communicative) via the senses” [25]. The body of literature on the study of perceptions of risk contains various perspectives, of which a number are discussed in this section.

How people perceive risks can be characterised by two dimensions, which follow the psychometric paradigm [26]. These two dimensions incorporate various psychometric characteristics of risk and are labelled ‘unknown’ and ‘dread/catastrophic potential’. The dimension ‘unknown’ covers characteristics such as ‘the risk is unknown to the exposed’, ‘the risk is not yet well known to science’, and ‘the risk has delayed effects’. The dread dimension covers, among others, characteristics as ‘the risk can have fatal consequences’, ‘the risk is involuntary’, ‘the risk is uncontrollable’ and ‘the risk is especially high for future generations’.

Many modern risks such as those related to rubber granulate can be characterised by these psychometric characteristics. The television programme which put rubber granulate on the public agenda also addressed various elements that fit into the psychometric paradigm in framing the risk issue: the risk would be largely unknown to science, parents were not aware of the potential risks posed by rubber granulate while their children were exposed, exposure to rubber granulate might have delayed and potentially deadly consequences (cancer) for future generations (children), and exposure to rubber granulate would be difficult to control while practicing sports on fields which contain it.

The psychometric paradigm explains why certain risks trigger a strong public response. However, to gain a more inclusive understanding of how people perceive risks, it is important to study common knowledge and beliefs regarding the risk among different people. Together, this knowledge and these beliefs form an intertwined framework of thoughts, also called a mental model of risk. Commonly reported are knowledge and beliefs regarding the hazardous properties of a hazard, the exposure to the hazard, the consequences of this exposure, and any possible risk mitigation [5,27,28,29,30].

Besides from understanding the ‘cognitive’ knowledge and beliefs in the mental models of lay people, the emotional or affective response to a risk has also been emphasised in studies on perceptions of risk [31,32]. People do not always consider all information about a risk carefully, but often trust their feelings. It has also been shown that peoples’ feelings towards risks can differ substantially from peoples’ knowledge and beliefs about these risks. In addition, these cognitive and affective responses can have different effects on behaviour [31]. It is therefore important to consider not only the knowledge and beliefs of risks, but also the feelings that people experience when thinking of these risks.

Perceptions of risk, both cognitive and affective, can change over time and are influenced by new information. In some instances, public perceptions of risk seem to change dramatically in a short time period. An important process in this rapid evolvement of perceptions of risk is the social amplification and attenuation of risks. Kasperson and colleagues have theorised how certain hazards, and information about these hazards, can influence the concerns of many people in society through interaction with “[…] psychological, social, institutional, and cultural processes in ways that may amplify or attenuate public responses to the risk or risk event” [33]. Heightened media attention seems to play an important role in the amplification and attenuation of perceptions of risks in society.

Perceptions of risk do not only change within persons; they also differ from one person to another. Factors indicating personal risk relevance were found to be especially strong indicators for variation in perceived risks between people. Previous studies have shown higher perceived risks among people with previous (negative) experiences of the risk [10,34], spatial proximity to the risk source [35,36] and exposure to a hazard of oneself or loved ones [37,38]. 

### 1.4. Research Questions

This paper discusses public perceptions of the alleged health risk posed by practicing sports on fields containing rubber granulate infill in the Netherlands, including the knowledge and beliefs and affective response, and this paper discusses the extent to which people prefer mitigation of the rubber granulate risk. We explored changes in perceptions of risk and mitigation preferences over time in a period with dense media-coverage of the topic, and the influence that new risk information following a risk assessment has on these perceptions of risk and mitigation preferences. We studied these perceptions of risk and mitigation preferences in groups with, and without, direct personal risk relevance, namely in people with, and without, offspring exposed to rubber granulate.

The following research questions are central to this paper:
How is the risk posed by rubber granulate perceived by the Dutch public, in terms of their knowledge and beliefs and affective response, and to what extent are people in favour of mitigating the risk posed by rubber granulate?Do perceptions of the risk posed by rubber granulate and mitigation preferences change over time?Are the changes in the perceptions of the risk posed by rubber granulate and mitigation preferences influenced by familiarity with new information about the risk posed by rubber granulate following a risk assessment study?Do perceptions of the risk posed by rubber granulate and preferences for mitigation differ between people with and without offspring exposed to rubber granulate?

## 2. Materials and Methods

### 2.1. The Case Study: The Risk Posed by Practising Sports on Fields Containing Rubber Granulate Infill

On 5 October 2016, a Dutch weekly television programme, specialising in critical documentaries about societal affairs, launched an episode about the potential health risks posed by rubber granulate. In the broadcast, rubber granulate was framed as a potentially severe health risk for people, especially children, who practice sports on synthetic turf fields containing rubber granulate because of carcinogenic chemicals found in the rubber. In the weeks that followed, other media in the Netherlands expanded on the suggested association between playing sport on fields containing rubber granulate infill and developing cancer. Consequently, considerable societal response followed, including immediate restrictions in the use of synthetic turf fields imposed by some sport clubs and municipalities.

Within a week of the broadcast, the Minister of Health Welfare and Sports asked the National Institute for Public Health and the Environment (RIVM) to conduct a study before the end of 2016 to provide clarity about the health risks of practicing sports on fields containing rubber granulate infill. Following extensive research, on 17 December 2016, RIVM concluded that the health risk by exposure to chemicals in rubber granulate is “virtually negligible”, and that it is, therefore, safe to practice sports on fields that contain rubber granulate infill [14]. See Box 2 for a summary of the results of the RIVM report.

Box 2Lay summary of the RIVM report on health 
risks posed by rubber granulate (translated from Dutch and slightly abbreviated for this paper). Source: www.rivm.nl.
**Research by RIVM shows that practicing sports on fields containing 
rubber granulate infill is safe. Rubber granulate contains numerous substances, 
but these substances are released from the rubber granulate in very small 
quantities. This is because the substances are more or less ‘enclosed’ in the 
rubber granulate. Because of this, the harmful effect to health is virtually 
negligible.**

**Practicing sports on fields containing rubber granulate.**
Rubber granulate contains numerous substances such as 
polycyclic aromatic hydrocarbons (PAHs), bisphenol A (BPA), phthalates 
(plasticisers), metals and benzothiazoles (including 2-MBT). There is little 
variation in concentrations of substances between different fields and 
different point of measurements at the fields. The results are therefore 
representative for all fields containing SBR rubber granulate in the 
Netherlands.
**No association with leukaemia**
There were no indications found of a relationship between 
practicing sports on fields containing rubber granulate and developing 
leukaemia or lymphoma in the available scientific literature. The composition 
of rubber granulate shows that chemicals that could cause leukaemia or lymphoma 
are not present (benzene and 1,3-butadiene) or are present in a very low amount 
(2-mercaptobenzothiazole). Since the late 1980s, a slight rise has been observed 
in the number of people aged between 10 and 29 years who developed leukaemia or 
lymphoma. This trend has not changed since pitches made of synthetic turf were 
first used in the Netherlands in 2001.
**Recommendation to adjust the requirements**
RIVM recommends adjusting the current requirements for rubber 
granulate, to those set for consumers products. The current requirements 
applying on rubber granulate are the so-called requirements for mixtures. The 
requirements for consumer products are more stringent: far lower amounts (100 
to 1000 times lower) of polycyclic aromatic hydrocarbons (PAHs) are allowed 
compared to the requirements for mixtures. The amount of PAH in rubber 
granulate is slightly higher than the standard set for consumer products. The 
European Chemicals Agency (ECHA) is currently conducting research to determine 
suitable requirements for rubber granulate.

### 2.2. Study Population and Procedure

Two surveys were conducted using an online survey panel (Flycatcher Internet Research, ISO 26362; panel members receive points for completion of online questionnaires, which can be exchanged for gift cards or charity donations). The first survey was carried out between 5 December, 2016 and 12 December 2016 (hereafter referred to as T1) and the second survey was administered between 27 January 2017 and 2 February 2017 (hereafter referred to as T2). The survey panel had, at the time of this study, an active panel population of approximately 10,000 Dutch residents, 1337 of whom were invited to participate in the first survey by e-mail. Fifty percent of these panel members were stratified by age, sex, level of education and provincial residence to represent the Dutch population aged 18 years and older. The other 50% of invited panel members were parents of a child aged 18 years or younger, of whom 35% were parent to a child who played soccer at the time the pre-selection was made. All the participants from the first survey were invited to participate in the second survey.

Three groups were included in our analyses: (1) parents of children who practiced sports on fields containing rubber granulate, (2) parents of children who did not practice sports on fields containing rubber granulate, and (3) individuals without children. Parents of children who practiced sports on fields containing rubber granulate, i.e., parents of exposed children (from now on referred to as ‘parents (E)’) are the group of parents of children exposed to rubber granulate. We compared this group with the parents of children who did not practice sports on fields containing rubber granulate (parents of non-exposed children, from now on referred to as ‘parents (NE)’) and with individuals with no children (from now on referred to as ‘individuals (NC)’).

Panel members provided active consent for participating in surveys and for the gathering and sharing of their demographic information when they joined the panel. Panel members were informed about the content and purpose of participating in our surveys before taking part in them. Participation took approximately 15 to 20 minutes per survey. The Clinical Expertise Centre RIVM reviewed the research protocol and determined that this research was not subject to the Dutch law for medical research involving human subjects (WMO) [39], and therefore concluded that it was exempt from seeking further approval from the Ethical Research Committee.

### 2.3. Measures

Survey questions addressed perceptions of risk, in terms of the participants’ knowledge and beliefs about rubber granulate (following a mental model conceptualisation) and negative affective response to rubber granulate. The survey questions also addressed the participants’ preferences for mitigating the risks posed by rubber granulate.

Knowledge and beliefs were operationalised as knowledge and beliefs about the type and amount of hazardous substances in rubber granulate (nature of the hazard); about the ways in which, and the extent to which, children were exposed to rubber granulate during sports on fields containing rubber granulate (exposure); and about potential health effects and the perceived probability that these health effects would occur (probability of health effects). To study negative affective response, we assessed anxious, worried and angry feelings towards rubber granulate. Preferences for measures to be adopted to mitigate the health risk posed by rubber granulate were operationalized as strict measures (strict mitigation preferences) and less strict, regulatory, measures (regulatory mitigation preferences). See Table 1 for a full overview of the measures.

At the end of the T2 survey, following the questions on perceptions of risk and mitigation preferences, respondents were shown a summary of the results from the RIVM report (see Box 2) and asked the following question: “prior to reading this text, were you already familiar with the research results from the RIVM?”, to which they could answer ‘yes’, ‘no’ and ‘I don’t know’ (in the analysis, the answer category ‘I don’t know’ was merged with the answer category ‘no’).

### 2.4. Analyses

Principal component analyses (PCA) and reliability analyses were performed to develop constructs for perceptions of risk and mitigation preferences, including multiple survey questions and/or items. Constructs were considered valid if the PCA indicated that the items within a construct load on the same component, and when the internal consistency indicated by Cronbach alpha was 0.7 or higher. Following these analyses, constructs were developed by averaging the responses to the survey questions and/or items. Table 1 shows the constructs, the corresponding survey questions, items and answer categories, and the Cronbach’s Alphas which resulted from the reliability analyses.

First, for both surveys, descriptive analyses were performed for each construct (nature of the hazard, exposure, potential health effects, probability of health effects, (negative) affective response, regulatory mitigation preferences, and strict mitigation preferences), within each group (parents of exposed children: parents (E); parents on non-exposed children: parents (NE); and individuals without children: individuals (NC)). Second, Pearson’s correlations (two-tailed) were calculated between all perception and mitigation preference constructs at T1 and T2. Third, unstructured linear multilevel analyses were performed to study changes over time, the effect of being familiar with the RIVM results on changes over time, and the differences between the three groups. Two models were constructed for all perception constructs and mitigation preference constructs as dependent variables. The first model included main effects of dummy variables for a group (with parents (E) as reference group) and a main effect for time (T1 = 0; T2 = 1). The second model was an extended version of the first, with a main effect for being familiar with the RIVM results, and an interaction effect of time and being familiar with the RIVM results added. All multilevel analyses controlled for age (dummy variables for quantiles), sex, and education (dummy variables for low, intermediate and high education).

## 3. Results

### 3.1. The Study Population

The first survey had a response rate of 77% (N = 1033) and the second survey had a response rate of 76% (N = 782). Table 2 shows a description of the study population at T1 per population group (i.e., parents(E) N = 240; parents (NE) N = 321; and individuals (NC) N = 472), in terms of gender, age and education level.

### 3.2. Perceptions of the Risk Posed by Rubber Granulate and Preferences for Mitigation at T1 and T2

Table 3 shows the means and standard deviations of perceptions of risk (nature of the hazard, exposure, potential health effects, probability of health effects, and negative affective response) and preferences for mitigation (regulatory mitigation preferences and strict mitigation preferences) among parents (E), parents (NE), and individuals (NC), at T1 and T2. At T1 the means of all constructs exceed the scale-median (3) to the high side of the scales, except for perceptions of the probability of health effects among individuals (NC). Notable are the considerable scores on preferences for mitigation preferences, with regard to strict measures, but especially with regard to regulatory measures. Pearson’s correlations (two-tailed) between the dependent variables vary between 0.39 and 0.75 (the correlations can be found in the Appendix A).

### 3.3. Changes in Perceptions of Risk and Mitigation Preferences Over Time

Table 4 (Model 1) shows changes in perceptions of risk and mitigation preferences over time. A decrease over time is observed for all measured perceptions of risk and mitigation preferences, with coefficients ranging from −0.11 (95% CI = −0.15/−0.06) for exposure to −0.41 (95% CI = −0.47/−0.35) for negative affective response. The strongest decreases were observed in probability of the health effects (β = −0.35, 95% CI = −0.42/−0.28), negative affective response (β = −0.41, 95% CI = −0.47/−0.35) and strict mitigation measures (β = −0.33, 95% CI = −0.40/−0.27).

### 3.4. The Influence of Being Familiar with the RIVM Results on Perceptions of Risk and Mitigation Preferences

Forty-two percent of the respondents indicated that they were familiar with the results of the RIVM report before filling in the survey at T2. In those who indicated that they were familiar with the RIVM results, we observed a stronger decrease over time in perceived risk and in the preferences for strict mitigation measures (see Table 4, Model 2). Only the decrease of regulatory mitigation preferences did not seem to be strengthened by familiarity with the RIVM results (β = −0.08, 95% CI = −0.19/0.02). The strongest interaction effects of time and familiarity with the results were observed for perceptions of the potential health effects (β = −0.29, 95% CI = −0.41/−0.17), the probability of these health effects (β = −0.34, 95% CI = −0.48/−0.20), and for strict mitigation preferences (β = −0.27, 95% CI = −0.40/−0.14).

### 3.5. Differences in Perceptions of the Risk Posed by Rubber Granulate and Preferences for Mitigation Between Population Groups

The results in Table 4 (Model 1 and 2) indicate higher perceived risk and stronger preferences of mitigation among parents (E) compared to individuals (NC). Observed differences between these groups (individuals (NC) compared to parents (E), based on model 2) were strongest for probability of health effects (β = −0.27, 95% CI = −0.47/−0.07), negative affective response (β = −0.36, 95% CI = −0.55/−0.16) and regulatory mitigation preferences (β= −0.31, 95% CI = −0.45/−0.16). The differences between these groups seem less strong with regard to perceptions of the exposure to rubber granulate (β = −0.14, 95% CI = −0.28/−0.01) and the potential health effects (β = −0.15, 95% CI = −0.31/0.01).

Parents (E) and parents (NE) do not differ as strongly in perceptions of risk and preferences for mitigation as perceptions and preferences of parents (E) differ from those of individuals (NC). Parents (NE) compared to parents (E) scored less on the constructs nature of the hazard (β = −0.20, 95% CI= −0.36/−0.04), negative affective response (β = −0.27, 95% CI = −0.45/−0.09), and regulatory mitigation preferences (β = −0.21, 95% CI = −0.35/−0.07). Differences between parents (NE) and parents (E) seem negligible with regard to perceptions of exposure, potential health effects and probability of these health effects, and with regard to preferences for strict mitigation measures. 

Additional analyses were performed to explore whether the groups changed differently in their perceived risk and mitigation preferences over time (by adding an interaction effect of time and group to the multilevel analyses) and by familiarity with the RIVM report (by adding a three-way interaction of time, familiarity with the RIVM results, and group). None of these interaction effects were found to be significant, indicating that there were no relevant differences between groups in regard their change in perceived risk and mitigation preferences over time and under influence of familiarity with the RIVM report.

## 4. Discussion

This study explored perceptions of the risks posed by practicing sports on fields containing rubber granulate and preferences for mitigation of this risk at two points in time among groups with, and without, offspring exposed to rubber granulate. Our results showed that perceived risk and preferences for mitigation decreased over time, and perceived risk and mitigation preferences decreased more among those who were familiar with new risk information following an RIVM risk assessment. Parents of children practicing sports on fields containing rubber granulate perceived the risks posed by rubber granulate as higher and had stronger preferences for mitigation of the risk posed by rubber granulate than people without offspring exposed to rubber granulate. In this section, we discuss our findings and their interpretation. This is followed by a short reflection on the authorities’ responses to public concerns about risk and the strengths and limitations of our study.

### 4.1. How Was the Risk Posed by Rubber Granulate Perceived by the Dutch Public and to What Extent Were People in Favour of Mitigation of This Risk?

Our first survey (T1) took place at a point in time characterised by considerable uncertainty and contrasting opinion in the public debate regarding the potential health risk posed by rubber granulate. We also found a variety in perceived risk and preferences for mitigation among respondents at T1. However, in general, people seemed to think that rubber granulate contains various hazardous substances, that children are substantially exposed to rubber granulate, that developing cancer because of exposure to rubber granulate is possible and even that children developing cancer because of rubber granulate is quite probable. In addition, people seemed to experience considerable negative affect when thinking of rubber granulate and they preferred mitigation of the risk, especially with regard to regulatory mitigation measures (more rigid requirements for hazardous substances and increased surveillance of safety with regard to products such as rubber granulate).

Overall, the perceived threat of exposure to rubber granulate by the public at T1 seemed in contrast to the message from RIVM that practicing sports on fields containing rubber granulate is safe [14]. Previous studies on perceptions of contemporary man-made risks also found differences in risk assessments by the public and by scientific experts [4,5,6]. Although our findings seem to indicate that the Dutch public perceived the threat posed by rubber granulate as considerable, we need to be cautious when making statements about the levels of risk perceptions in terms of high and low perceived risk, as we do not have a real baseline measurement (prior to the television programme which put rubber granulate on the public agenda).

### 4.2. Did Perceptions of Risk and Mitigation Preferences Related to Rubber Granulate Change Over Time, and Were These Changes in Perceptions and Preferences Influenced by Familiarity With New Information About the Risk Posed by Rubber Granulate Following Publication of a Risk Assessment Study?

Between our first and second survey, RIVM published a report, after an extensive risk assessment study, stating that practicing sports on fields containing rubber granulate is safe. Our results indicate a decrease in the perceived risk posed by rubber granulate and preferences for mitigation in the period between the first and second survey. The largest decreases were observed in the perceived probability of health effects caused by rubber granulate, the negative affective responses to rubber granulate, and preferences for strict mitigation measures of the risk posed by rubber granulate.

The decrease in all the perceptions of risk, and in preferences for regulatory mitigation, was stronger for those who were familiar with the results off the RIVM report. Three implications can be drawn from the influence of being familiar with the RIVM report on the decline in perceived risk and mitigation preferences. First, the effect of being familiar with the RIVM information was most pronounced in relation to perceptions of the potential health effects, the probability of these health effects, and for strict mitigation preferences. The stronger decline of these perceptions seems to indicate that the RIVM’s message, “practicing sports on fields containing rubber granulate infill is safe” (see Box 2), has, at least to some extent, been understood and accepted by those who read or heard about the results off the RIVM report. Second, preferences for regulatory mitigation declined over time, but do not appear to have been influenced by the RIVM information in any significant way. This also seems quite in line with the RIVM recommendation to adjust current EU requirements for polycyclic aromatic hydrocarbons (PAHs) in rubber granulate (see Box 2). Third, while negative affective response to rubber granulate decreased the most of all perceptions of risk over time, familiarity with the RIVM results had a stronger effect on the decline in perceptions of the potential health effects and the probability of these health effects than it had on negative feelings towards rubber granulate (e.g., “it does not feel right”). Previous research has emphasized that affective or emotional responses to (information about) risks can differ significantly from cognitive responses to this risk [31]. Importantly, Loewenstein and colleagues have also emphasized that when these divergent cognitive and affective responses occur, the affective responses likely have the strongest impact on behaviour. Therefore, a recommendation for future communications about risk might be to anticipate the feelings of the recipients more accurately, as anticipating feelings might have a considerable impact in health and risk communication, especially among those with personal risk relevance [41].

The RIVM risk assessment report was not the only factor influencing the decrease in perceived risk and mitigation preferences. Although perceptions of risk and mitigation preferences dropped most significantly in people who were familiar with the results of the RIVM report, we also saw a significant decline in perceptions of risk and mitigation preferences in those who indicated that they were unfamiliar with the results. The decline in perceptions of risk and mitigation preferences could be related to the decline in media coverage of the topic over time. While rubber granulate was a recurrent topic in the media during the period of data collection, by far the highest peak of media attention was in the weeks directly after the television programme suggesting that there was a potential health risk posed by rubber granulate. The relationship between media attention and perceptions of risk has been explored in previous studies. A literature review by Wahlberg and Sjoberg shows that current studies mainly provide evidence off a relationship between the amount of media attention about a risk topic and risk perceptions [42]. The amount of media-attention, irrespective of its content, is shown to be associated with heightened perceived risk. This relationship between the quantity of media-attention and perceived risk can be explained by the availability-heuristic. Hence, following the availability-heuristic, the mental representations that are easily available in people’s minds are assessed as being more probable than events that are not easily mentally accessible. A high amount of media-attention on a risk topic increases the mental availability and, therefore, the perceived probability of the risk [42]. The peak in media attention at the start of the rubber granulate affair was probably better remembered by respondents at T1 than at T2, leading to a higher mental availability at T1 than at T2, potentially explaining the decrease of perceived risk and the preferences for mitigation between T1 and T2, irrespective of their familiarity with the results off the RIVM report.

### 4.3. Do Perceptions of the Risk Posed by Rubber Granulate and Preferences for Mitigation Differ Between People With, and Without, Offspring Exposed to Rubber Granulate?

Parents of children who practiced sports on fields containing rubber granulate perceived the risk posed by rubber granulate as higher than individuals without children, and they had stronger preferences for mitigation of the risk. This finding was expected as previous literature has shown higher perceived risks among people with previous risk experiences [10,34], spatial proximity to the risk source [36,37] and exposure to the hazard by oneself or loved ones [37,38].

Comparing parents of exposed children with parents of non-exposed children, parents of exposed children seemed more convinced about the presence of hazardous substances in rubber granulate, experienced more negative affect when thinking of rubber granulate and were more in favour of regulatory mitigation measures. Nevertheless, we did not observe any relevant differences in perceptions of exposure, potential health effects and probability of health effects, nor in preferences for strict mitigation measures. It seems that simply being a parent, irrespective of having a child who is exposed to rubber granulate, is related to a higher perceived risk and stronger mitigation preferences. Previous research has also suggested that parents generally perceive risks as higher than non-parents [43,44], and more strongly support risk mitigation measures [45]. A possible reason for this association is that parents experience heightened levels of caution for risks in general, because they are caretakers of vulnerable beings. The study by Cameron et al. also supports this explanation as the authors found that risk aversion was higher among parents of infants than it was among parents of teenagers [45]. An infant, after all, is more likely to be perceived as a vulnerable being than a teenager is.

### 4.4. Authorities’ Responses to Concerns About Risks

Concerns about rubber granulate on sports fields have not been restricted to the Netherlands; concerns have also been raised in the United States and in various countries in Europe [16,46]. This has led to several literature reviews and additional risk assessments being carried out by public health institutes [11,12,13,15,16]. The vast majority of these studies support the general conclusion that there are no indications off a relevant health risk caused by practicing sports on fields containing rubber granulate infill. Whether more studies on the safety of rubber granulate will reassure concerned citizens is, however, unclear. Although our study shows changes in perceptions of risk over time and after receiving information from a risk assessment, the perceptions and mitigation preferences recorded at T2 do not suggest that everyone is reassured about the risk posed by rubber granulate. The application of risk communication knowledge in practice can help bridge the concerns from citizens and the conclusions following risk scientific studies [22]. A main priority should be to formulate risk communication, which is adapted to the perceptions of risk in various groups to support decision making. Risk communication should address decision relevant misconceptions, complement knowledge gaps, and address the particular elements that a particular audience considers important. At this point, a better understanding of the perceptions of risk posed by rubber granulate might arguably be of higher importance than conducting more risk assessment studies.

### 4.5. Strengths and Limitations

This paper is, to our knowledge, the first to address perceptions of risk and mitigation preferences regarding rubber granulate on sport fields. The data for this study was collected during an ongoing public discussion among various population groups and was therefore successful at exploring changes in perceptions of risk and mitigation preferences under the influence of changing social dynamics, and in identifying differences among groups which had, or did not have, direct risk relevance. Our study also has a number of limitations. First, our study was conducted exclusively by means of surveys. A mixed methods approach, including exploratory qualitative research followed by confirmatory surveys, would arguably be the best way to get a complete and accurate of the public perceptions of risks [47]; however, in an attempt to cover changes in perceptions in real-time dynamic discussions, a full mixed methods study can be challenging or even unfeasible because of time constraints. For future research, we advise investing in the development of mixed-method approaches for studying perceived risks in risk events or dynamic discussions of risk in a relatively short period of time. Another limitation is that our study lacks a clear baseline measurement for studying changes in perceptions of risk and mitigation measures during the public debate on rubber granulate. A baseline measurement, conducted at a moment of limited media attention, would have enabled us to make explicit statements about the levels of perceptions of risk and mitigation preferences at T1 and T2. In addition, it would have aided us in interpreting the role of media attention on perceptions of risk and mitigation preferences. However, considering the unexpected nature of this public debate, a baseline measurement was hardly possible.

## 5. Conclusions

This paper addressed public perceptions of the risk posed by practicing sports on fields containing rubber granulate and preferences for mitigation of this risk, against the background of ongoing public debate about this risk in the Netherlands. The results of this study show that, initially, a substantial proportion of the Dutch public perceived practicing sports on fields containing rubber granulate as a potential health threat to children. Over time, after RIVM published a new risk assessment study stating that practicing sports on fields containing rubber granulate is safe, the perceived risk and preferences for mitigation of this risk decreased, especially among those who were familiar with the new information. Parents of children under the age of 18, in particular those with children who were exposed to rubber granulate, were more likely to perceive the risk as higher and to prefer a stricter mitigation policy.

New information about health risks released by public health institutes can reassure the public and reduce concerns about risks. Nevertheless, other factors influencing perceptions of risk and mitigation preferences over time should not be forgotten (e.g., media attention). In communicating information about risks, the differences between people based on personal risk relevance, and perhaps also the presence or absence of parenthood, should be taken into account. More insights into detailed perceptions of risk among various groups in society, and the influence of new information and other societal factors affecting these perceptions, could provide public health institutions with focal points for their future communication about risks.

## Figures and Tables

**Table 1 ijerph-16-02250-t001:** Operationalization of dependent variables. The names of the constructs, the corresponding survey questions, items and answer categories, and the Cronbach’s Alphas which resulted from the reliability analyses are presented here.

Construct	Question(s)	Items	Answer Categories	Scale’s Cronbach’s Alpha (Survey) *
Perceptions of the nature of the hazard	What substances are present in rubber granulate in your opinion?	Carcinogenic substances Poisonous substances	Five-point Likert scale: 1. Certainly not … 5. Certainly yes	0.87 (T1) 0.89 (T2)
Perceptions of the exposure to rubber granulate	How often do you think a child comes into contact containing rubber granulate when practicing sports on fields containing rubber granulate?	Contact of rubber granulate with the bare skin Contact of rubber granulate with wounds or small wounds Rubber granulate getting into the mouth Swallowing rubber granulate Inhalation of fumes of rubber granulate	Five-point Likert scale: Never Rarely Sometimes Frequently Always I do not know **	0.82 (T1) 0.85 (T2)
Perceptions of the potential health effects	In your opinion, could the following health issues be caused by practicing sports on fields containing rubber granulate?	Health issues such as headache, nausea, rash, pain in muscles and joints Cancer	Five-point Likert scale: 1. Certainly not … 5. Certainly yes	0.78 (T1) 0.86 (T2)
Perceptions of the probability of health effects	In your opinion, what is the chance of children developing health issues such as headache, nausea, rash, pain in muscles and joints due to practicing sports on fields of rubber granulate for children? In your opinion, what is the chance of children getting cancer due to practicing sports on fields of rubber granulate?		Five-point Likert scale: 1. Small chance … 5. Large chance	0.86 (T1) 0.90 (T2)
(Negative) affective response to the risk	What are your feelings when you think of the use of rubber granulate on artificial fields?	Not anxious - anxious Not angry - angry Not worried - worried	Five-point semantic scale, e.g., 1. Not anxious … 5. Anxious	0.86 (T1) 0.90 (T2)
Preferences for mitigation (regulatory)	What do you think of the use of rubber granulate on artificial sport fields? Please indicate to what extent you agree or disagree with the following statements:	Standards for hazardous substances in products such as rubber granulate should become more strict There should be more surveillance on the safety of products such as rubber granulate	Five-point Likert scale: Do not agree at all Do not agree No opinion Do agree Do very much agree	0.82 (T1) 0.83 (T2)
Preferences for mitigation (strict)	In your opinion, should there be steps taken at this moment? Please indicate to what extent you agree or disagree with the following statements:	No more practicing sports on turf fields containing rubber granulate No more young children practicing sports on turf fields containing rubber granulate	Five-point Likert scale: Do not agree at all Do not agree No opinion Do agree Do very much agree	0.90 (T1) 0.90 (T2)

* Scales were constructed by averaging the responses to the total number of items; ** the option ‘I do not know’ was coded in the analysis as system missing, leading to a 2.7% non-response to the scale ‘exposure to rubber granulate’.

**Table 2 ijerph-16-02250-t002:** The study population at T1: gender, age (mean and standard deviation) and education level per population group.

	Parents (E) *	Parents (NE)	Individuals (NC)	Total
Women—N (%)	142 (59.2)	189 (58.9)	219 (46.4)	550 (53.2)
Age in years—M (sd)	44.5 (6.4)	43.2 (7.6)	50.5 (17.6)	46.8 (13.4)
Education level **				
LowN (%)	23 (9.6)	43 (13.4)	165 (35.0)	231 (22.4)
Intermediate—N (%)	96 (40.4)	121 (37.7)	199 (42.2)	417 (40.4)
High—N (%)	120 (50.0)	157 (48.9)	108 (22.9)	385 (37.3)
Total—N (%)	240 (23.2)	321 (31.1)	472 (45.7)	1033 (100)

* Parents (E): parents of exposed children; parents (NE): parents of non-exposed children; and individuals (NC): individuals without children; ** following the standard operationalisation of the Dutch Central Bureau of Statistics [40]: ‘low’ includes up to primary education, secondary education preparing for vocational education, and first-phase secondary education preparing for higher education; ‘intermediate’ includes up to secondary education preparing for higher education and vocational education; ‘high’ includes higher education and post-university education (this is an abbreviated operationalization, see abovementioned reference of CBS for the full operationalization of education level).

**Table 3 ijerph-16-02250-t003:** Means and standard deviations of perceptions of risks related to rubber granulate and preferences for mitigation per group at T1 and T2 *.

	Parents (E) **		Parents (NE) **		Individuals (NC) **	
	T1	T2	T1	T2	T1	T2
	M (sd)	M (sd)	M (sd)	M (sd)	M (sd)	M (sd)
Nature of the hazard	3.8 (0.9)	3.6 (0.9)	3.6 (0.8)	3.4 (1.0)	3.6 (0.9)	3.3 (0.9)
Exposure	3.3 (0.7)	3.2 (0.7)	3.3 (0.7)	3.2 (0.7)	3.2 (0.7)	3.1 (0.7)
Potential health effects	3.2 (0.9)	3.0 (0.9)	3.2 (0.8)	2.9 (0.9)	3.1 (0.9)	2.9 (1.0)
Probability of health effects	3.0 (1.1)	2.8 (1.2)	3.0 (0.9)	2.6 (1.1)	2.9 (1.1)	2.6 (1.2)
Negative affective response	3.4 (1.1)	2.9 (1.0)	3.1 (1.0)	2.7 (1.0)	3.1 (1.0)	2.7 (1.0)
Regulatory mitigation preferences	4.3 (0.7)	4.1 (0.9)	4.1 (0.8)	3.9 (0.8)	4.1 (0.8)	3.8 (0.8)
Strict mitigation preferences	3.6 (1.2)	3.3 (1.2)	3.5 (1.0)	3.2 (1.1)	3.5 (1.0)	3.1 (1.1)

* N parents (E): 240 (T1) and 180 (T2); N parents (NE): 321 (T1) and 236 (T2); N individuals (NC): 472 (T1) and 365 (T2). The sample size for the variable exposure is lower than the sample size for the other variables, due to the coding of the answer category ‘I do not know’ as missing values (see methods section). N for exposure in parents (E): 239 (T1) and 181 (T2); N for exposure in parents (NE): 310 (T1) and 232 (T2); N for exposure in individuals (NC): 456 (T1) and 360 (T2). ** Parents (E): parents of exposed children; parents (NE): parents of non-exposed children; individuals (NC): individuals without children.

**Table 4 ijerph-16-02250-t004:** Differences in perceptions of the risk posed by rubber granulate and mitigation preferences between population groups *, changes in perceptions and preferences over time and the effect of being familiar with the RIVM results on the changes in perceptions and preferences over time; the results of multilevel analyses.

		Model 1 **				Model 2 **			
		*β*	*p*-value	95% CI		*β*	*p*-value	95% CI	
Hazard	Individual (NC) ***	−0.21	0.015	−0.38	−0.04	−0.21	0.015	−0.38	−0.04
	Parents (NE) ***	−0.20	0.015	−0.36	−0.04	−0.20	0.017	−0.36	−0.04
	Time	−0.28	0.000	−0.34	−0.22	−0.22	0.000	−0.30	−0.14
	Familiarity results					0.10	0.133	−0.03	0.24
	Time * Familiarity results					−0.13	0.035	−0.26	−0.01
Exposure	Individual (NC) ***	−0.14	0.035	−0.28	−0.01	−0.14	0.035	−0.28	−0.01
	Parents (NE) ***	−0.03	0.666	−0.15	0.10	−0.02	0.708	−0.15	0.10
	Time	−0.11	0.000	−0.15	−0.06	−0.07	0.037	−0.13	0.00
	Familiarity results					0.10	0.066	−0.01	0.20
	Time * Familiarity results					−0.10	0.039	−0.19	0.00
Potential health effects	Individual (NC) ***	−0.15	0.067	−0.31	0.01	−0.15	0.066	−0.31	0.01
	Parents (NE) ***	−0.06	0.450	−0.21	0.09	−0.07	0.384	−0.22	0.09
	Time	−0.26	0.000	−0.32	−0.20	−0.14	0.001	−0.21	−0.06
	Familiarity results					0.03	0.640	−0.10	0.16
	Time * Familiarity results					−0.29	0.000	−0.41	−0.17
Probability of health effects	Individual (NC) ***	−0.27	0.009	−0.47	−0.07	−0.27	0.009	−0.47	−0.07
	Parents (NE) ***	−0.08	0.420	−0.27	0.11	−0.09	0.363	−0.28	0.10
	Time	−0.35	0.000	−0.42	−0.28	−0.20	0.000	−0.29	−0.11
	Familiarity results					0.04	0.621	−0.12	0.20
	Time * Familiarity results					−0.34	0.000	−0.48	−0.20
Negative affective response	Individual (NC) ***	−0.36	0.000	−0.55	−0.16	−0.36	0.000	−0.55	−0.16
	Parents (NE) ***	−0.27	0.004	−0.45	−0.08	−0.27	0.004	−0.45	−0.09
	Time	−0.41	0.000	−0.47	−0.35	−0.31	0.000	−0.39	−0.24
	Familiarity results					0.05	0.484	−0.09	0.20
	Time * Familiarity results					−0.23	0.000	−0.34	−0.11
Regulatory mitigation preferences	Individual (NC) ***	−0.31	0.000	−0.45	−0.16	−0.31	0.000	−0.45	−0.16
	Parents (NE) ***	−0.21	0.002	−0.35	−0.08	−0.21	0.003	−0.35	−0.07
	Time	−0.23	0.000	−0.28	−0.18	−0.20	0.000	−0.26	−0.13
	Familiarity results					0.09	0.137	−0.03	0.20
	Time * Familiarity results					−0.08	0.108	−0.19	0.02
Strict mitigation preferences	Individual (NC) ***	−0.22	0.032	−0.42	−0.02	−0.22	0.032	−0.42	−0.02
	Parents (NE) ***	−0.07	0.478	−0.26	0.12	−0.07	0.468	−0.26	0.12
	Time	−0.33	0.000	−0.40	−0.27	−0.22	0.000	−0.30	−0.13
	Familiarity results					0.11	0.160	−0.04	0.27
	Time * Familiarity results					−0.27	0.000	−0.40	−0.14

* Parents (E) (parents of exposed children) is the reference group; ** Model 1: the influence of group and time on risk perceptions and mitigation preferences; Model 2: Model 1 extended with the main effect of being familiar with the RIVM results and the interaction effect of time and being familiar with the RIVM results; The analyses in both models are controlled for age (three dummy variables for quantiles), education (two dummy variables), and sex; *** Parents (NE): parents of non-exposed children, and individuals (NC): individuals without children.

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
