# Peer review of "Public Perceptions of Contentious Risk: The Case of Rubber Granulate in the Netherlands"

_ijerph, 2019, doi:10.3390/ijerph16122250_

Round 1

Reviewer 1 Report

The authors provided a manuscript examining the risk perceptions of practicing sports on fields with rubber granulate and the preferences for mitigation measures amongst people with and without children exposed to rubber granulate. The manuscript would be of interest to those who are interested in risk perception and environmental health, and it focuses on a topic that has been increasingly gaining attention.

The work presented is appropriate; however, there are grammatical errors that must be fixed throughout the paper (e.g., excessive use of commas throughout and other places where commas are missing). It also seems as though the paper would benefit from some restructuring such as providing more information in the introduction on what rubber granulate is and the potential health concerns of rubber granulate exposure (this is really only first introduced in the methods section).

There are also some additional comments for consideration throughout the paper. It seems like either perceptions of risk or risk perception should be chosen and used throughout the paper rather than going back and forth between the two. Another point is that in some places, T1 and T2 are referred to as surveys and in others they are referred to as measurements, so it is best to pick one form of wording and use that throughout. The formatting of quoted material is inconsistent and should be addressed (i.e., some passages are italicized and others are not and the in text citation formatting should be consistent). Spelling needs to be double checked for errors and also to ensure consistency in chosen spelling of words throughout (e.g., practicing vs practising, program vs programme). All values presented in the text should be double checked to ensure they match what is being presented in the tables (e.g., line 269 shows 0.11 but the table shows -0.11). There should be consistency in capitalization throughout the document (e.g., some sentences capitalize table within the text and others do not, some capitalize parents(E) and others do not). Finally, the references are not formatted consistently and should be edited.

More specific comments are as follows:

Line 27: Should include N = for second survey (similar to line 25)

Line 30: Change two moments to two time periods

Line 37: Remove ‘and’

Section 1: Would like to see more about rubber granulate background in introduction (what it is, where it comes from, what the potential health impacts are, etc.)

Line 61: Is there a more recent reference to support this?

Lines 61-64: Reword sentence for clarity

Line 68: e.g. not needed before the references. Also, what about nay references from EPA or CDC/ATSDR? Should elaborate on the findings from the references presented here to give the reader more background

Line 73: Uncertainties such as? Give more information

Line 74: Start new paragraph with Uncertainty is at the core…

Line 74: Reword – lay people feel on the average uncomfortable to lay people generally feel uncomfortable

Line 76: Change uncomfortability to discomfort

Line 77: Change to Thus, scientific risk assessments alone will likely not…

Line 81: Add bridge the gap between scientific assessments of risk…

Line 84: Change to In order to develop effective risk communication, and support the public….

Line 90: Change to How people perceive risk can be characterized by two dimensions, which follow the…

Line 92: Change were to are

Line 115: Change to but often trust based on their feelings

Lines 137-138: Change to …granulate infill in the Netherlands. This includes knowledge, beliefs, and affective response as well as the extent to which people prefer…

Line 155: Change to …weekly television program, specializing in…

Box 1: Change to …aged between 10 and 29 who got leukemia or lymphoma

Line 176: Change fast to vast

Line 181: Suggest adding something like (hereafter referred to as T1)

Line 183: Suggest changing to 1337 panel members instead of 1337 Panel

Lines 185-186: Change to The other 50% of invited panel members were a parent to …

Line 189: Suggest lowercase parents

Line 194: Change from to of

Line 205: Change from risk perceptions to risk perception

Line 217: Add ‘the’ after following

Line 225: Should read as Cronbach’s Alpha

Lines 226-229: Replace semicolons with commas

Lines 232-233: Change to …each risk perception using the mitigation preference construct as the dependent variable

Line 234: Change to …dummy variables for a group (with Parents(E) as reference group)…

Line 237: Remove the word were

Section 2.4 Analyses: This section does not seem to mention the analysis methods for what is presented in the Appendix

Table 1 heading: Change to Cronbach’s Alpha

Table 1: Should change to Five-point Likert scale throughout and also ensure consistency in formatting throughout table (e.g., Items are capitalized under preferences for mitigation but not capitalized for other sections, punctuation is used after preferences for mitigation strict but not in other sections)

Table 1 first row: Change to Scale Cronbach’s Alpha

Table 1 negative affective response row: Options presented under items (e.g., (not) anxious, etc.) are not clear – is it not anxious or is it anxious?

Table 1 notes: It would be clearer if each note was on a separate line

Lines 247-248: Change to i.e., parents(E) N = 240; parents(NE) N = 321; and individuals(NC) N = 472)

Table 2: Should specify what the M stands for. Median?

Table 2 notes: It would be clearer if each note was on a separate line. Should also clarify here what low, intermediate, and high mean.

Line 250: Change to Parents(NE): parents of non-exposed…

Line 252: Should capitalize Central Bureau of Statistics

Lines 255-256: Reword sentence for clarity

Table 3: It seems like it would be good to provide a bit more information in the text for the reader to know exactly what this table is showing us.

Table 3: Footnote 2 seems like it should be in the notes of the table rather than as a footnote

Line 280: Change to …health effects (β=-0.29, 95% CI=-0.41/-0.17), the probability of these…

Line 284: Remove comma after (Model 1 and 2)

Line 291: Remove ‘show to’

Lines 291-292: Reword sentence for clarity

Footnotes 2 and 3: Font is not consistent. Change to None of these interaction effects were found to be significant…

Table 4: It would be clearer if each note was on a separate line

Table 4, lines 302-303: Need a space between education and (two dummy variables). Add comma before ‘and sex’. Change to Parents(NE): parents of non-exposed children, and Individuals(NC)…

Line 309: Use wording other than ‘change stronger’

Line 316: Extra punctuation at start of heading

Lines 353-360: Is there some literature that can be included in this section to support the findings? Would be good to include even if it’s not specific to rubber granulate

Line 367-368: Change to …who indicated that they were unfamiliar…

Lines 372-373: The relationship between media and risk perceptions has been explored – and what has this shown? More references needed here and should elaborate on this a bit

Line 373: Reword quantity of media attention on itself can affect for clarity

Line 376: Check citation formatting punctuation

Lines 376-377: It seems like it would be better if this quote was paraphrased instead and given more context

Line 384: Add ‘they’ after and

Lines 396-398: Elaborate on the literature more to provide more information for the reader

Line 402: Why is this? One would think they’re serving as caregivers as well. Is this because infants are more vulnerable than teenagers? Add additional sentence after referring to Cameron 2010 study

Line 405: Are there references to include about the concerns raised elsewhere?

Line 407: Don’t need e.g., - just include the refs

Lines 415-417: Reword for clarity

Line 425: Change to Our study also has a…

Line 432: Change to …discussions of risk in a relatively short period of time

Line 446: Remove the word ‘and’

Line 451: Other factors such as?

Appendix: Why is this only shown for T1 and not for T2?

Author Response

Dear reviewer 1,

Thank you for your considerate comments. We have carefully considered all comments in preparing our revision. Hereby we send you our point-to-point response.

 -----------------------------------------------------------------------------------------------------------------------------

The authors provided a manuscript examining the risk perceptions of practicing sports on fields with rubber granulate and the preferences for mitigation measures amongst people with and without children exposed to rubber granulate. The manuscript would be of interest to those who are interested in risk perception and environmental health, and it focuses on a topic that has been increasingly gaining attention.

The work presented is appropriate; however, there are grammatical errors that must be fixed throughout the paper (e.g., excessive use of commas throughout and other places where commas are missing).

Response authors: A certified corrector has checked our manuscript on correct language use and grammar and we have adapted the manuscript accordingly.

It also seems as though the paper would benefit from some restructuring such as providing more information in the introduction on what rubber granulate is and the potential health concerns of rubber granulate exposure (this is really only first introduced in the methods section).

Response authors: We have added a Box in the introduction, clarifying origin, use and potential health risks of rubber granulate.

There are also some additional comments for consideration throughout the paper. It seems like either perceptions of risk or risk perception should be chosen and used throughout the paper rather than going back and forth between the two.

Response authors: We agree with the reviewer and have changed the terminology consistently throughout the paper into ‘perceptions of risk’ (except for the key words).

Another point is that in some places, T1 and T2 are referred to as surveys and in others they are referred to as measurements, so it is best to pick one form of wording and use that throughout.

Response authors: All references to T1 and T2 have been changed into ‘surveys’. 

The formatting of quoted material is inconsistent and should be addressed (i.e., some passages are italicized and others are not and the in text citation formatting should be consistent).

Response authors: We have rechecked the document on formatting of quoted material, and adapted the text accordingly.

Spelling needs to be double checked for errors and also to ensure consistency in chosen spelling of words throughout (e.g., practicing vs practising, program vs programme).

Response authors: A certified corrector has checked our manuscript on correct language use and grammar and we have adapted the manuscript accordingly.

All values presented in the text should be double checked to ensure they match what is being presented in the tables (e.g., line 269 shows 0.11 but the table shows -0.11).

Response authors: We have double checked all values in the manuscript and corrected the value identified by the reviewer.

There should be consistency in capitalization throughout the document (e.g., some sentences capitalize table within the text and others do not, some capitalize parents(E) and others do not). Finally, the references are not formatted consistently and should be edited.

 Response authors: The manuscript has been checked on consistent capitalization of groups (parents(E), parents(NE) and individuals(NC)), table references and box references.

More specific comments are as follows:

 Line 27: Should include N = for second survey (similar to line 25)

 Response authors: We have added the sample size of the second survey to the indicated line. 

Line 30: Change two moments to two time periods

Response authors: We have rephrased the sentece into the following: Multilevel analyses were performed to study changes in perceptions of risk and mitigation preferences in the time between the two surveys, the influence of being familiar with new information following the risk assessment report, and the differences in the perceptions of risk and mitigation preferences between groups with and without offspring exposed to rubber granulate.

Line 37: Remove ‘and’

Response authors: We have changed the text as suggested.

Section 1: Would like to see more about rubber granulate background in introduction (what it is, where it comes from, what the potential health impacts are, etc.)

Response authors: See our earlier response; we have added a Box to the introduction to clarify the origin, use and potential health risks of rubber granulate.

Line 61: Is there a more recent reference to support this?

Response authors: We have added a more recent reference: Sørensen, M. P. (2018). Ulrich Beck: exploring and contesting risk. Journal of Risk Research, 21(1), 6-16.

Lines 61-64: Reword sentence for clarity

Response authors: We have rephrased the sentence into the following two sentences: Public debates about these so called ‘modern risks’ are often characterised by concerns about invisible hazardous substances, and potential, though often highly uncertain, health consequences (Beck, 1992; Sørensen, 2018). These public concerns about risks can have considerable societal impact. Public concerns have been shown to be, although not univocal, associated with individual risk (mitigation) behaviour and support for risk management (Huddy, Feldman, Taber, and Lahav, 2005; Wachinger, Renn, Begg, and Kuhlicke, 2013).

 Line 68: e.g. not needed before the references. Also, what about nay references from EPA or CDC/ATSDR? Should elaborate on the findings from the references presented here to give the reader more background

Response authors: We have deleted ‘ e.g.’ and added the reference from the Federal research action plan on recycled tire crumb used on playing fields and playgrounds by EPA and other institutions. We have also added some additional information about the conclusion from the currently published studies on health risks of exposure to rubber granulate.

Line 73: Uncertainties such as? Give more information

Response authors: We clarified with the following text: The overall conclusion of these studies is that there are no serious health risks to be expected as a result of practicing sports on fields containing rubber granulate. Nevertheless, conclusions like this following a literature review or risk assessment always bring with them a certain level of uncertainty. This uncertainty is either because of study limitations (e.g. a limited sample size or a lack of longitudinal data) or simply because of scientific nuance (e.g. a common scientific phrase is ‘a risk cannot be excluded’) (Aven and Zio, 2011; Jansen, Claassen, van Poll, van Kamp, and Timmermans, 2017; Walker et al., 2003). 

Line 74: Start new paragraph with Uncertainty is at the core…

Response authors: We have changed the text as suggested.

Line 74: Reword – lay people feel on the average uncomfortable to lay people generally feel uncomfortable

Response authors: We have changed the text as suggested.

Line 76: Change uncomfortability to discomfort

Response authors: We have changed the text as suggested.

Line 77: Change to Thus, scientific risk assessments alone will likely not…

Response authors: We have rephrased the text into the following: Scientific risk assessments alone will therefore probably not adequately meet the public’s need for information and reassurance, and might even increase concerns and the demand for risk mitigation because of to their inherent uncertainties.

Line 81: Add bridge the gap between scientific assessments of risk…

Response authors: We have changed the text as suggested.

Line 84: Change to In order to develop effective risk communication, and support the public….

Response authors: We have changed the text as suggested.

Line 90: Change to How people perceive risk can be characterized by two dimensions, which follow the…

Response authors: We have changed the text as suggested.

Line 92: Change were to are

Response authors: We have changed the text as suggested.

Line 115: Change to but often trust based on their feelings

Response authors: We have rephrased the sentence, following the suggestion of the other reviewer, into: People do not always consider all information about a risk carefully, but often trust their feelings.

Lines 137-138: Change to …granulate infill in the Netherlands. This includes knowledge, beliefs, and affective response as well as the extent to which people prefer…

Response authors: We have rephrased the text for clarity into: This paper discusses public perceptions of the alleged health risk of practicing sports on fields with rubber granulate infill in the Netherlands, including knowledge and beliefs and affective response, and this paper discusses the extent to which people prefer mitigation of the rubber granulate risk.

Line 155: Change to …weekly television program, specializing in…

Response authors: We have changed the text as suggested.

Box 1: Change to …aged between 10 and 29 who got leukemia or lymphoma

Response authors: We have changed the text as suggested.

Line 176: Change fast to vast

Response authors: We have changed the text as suggested.

Line 181: Suggest adding something like (hereafter referred to as T1)

Response authors: We have changed the text as suggested.

Line 183: Suggest changing to 1337 panel members instead of 1337 Panel

Response authors: We have rephrased the text into the following: The survey panel had, at the time of this study, an active panel population of approximately 10,000 Dutch residents, 1337 of whom were invited to participate in the first survey by e-mail.

Lines 185-186: Change to The other 50% of invited panel members were a parent to …

Response authors: We have changed the text as suggested.

Line 189: Suggest lowercase parents

Response authors: We have changed the text as suggested.

Line 194: Change from to of

Response authors: We have changed the text as suggested.

Line 205: Change from risk perceptions to risk perception

Response authors We have changed the terminology consistently throughout the paper into ‘perceptions of risk’.

Line 217: Add ‘the’ after following

Response authors: We have changed the text as suggested.

Line 225: Should read as Cronbach’s Alpha

Response authors: We have changed the text as suggested.

Lines 226-229: Replace semicolons with commas

Response authors: We have changed the text as suggested.

Lines 232-233: Change to …each risk perception using the mitigation preference construct as the dependent variable

Response authors: We have rephrased the sentence for clarity into: Two models were constructed for all risk perception constructs and mitigation preference constructs as dependent variables. 

Line 234: Change to …dummy variables for a group (with Parents(E) as reference group)…

Response authors: We have changed the text as suggested.

Line 237: Remove the word were

Response authors: We have changed the text as suggested.

Section 2.4 Analyses: This section does not seem to mention the analysis methods for what is presented in the Appendix

Response authors: We have added the analysis methods for the correlations in the Appendix to section 2.4 Analysis, with the following text: […] Pearson’s correlations (two-tailed) were calculated between all perception and mitigation preference constructs at T1 and T2. 

Table 1 heading: Change to Cronbach’s Alpha

Response authors: We have changed the text as suggested.

Table 1: Should change to Five-point Likert scale throughout and also ensure consistency in formatting throughout table (e.g., Items are capitalized under preferences for mitigation but not capitalized for other sections, punctuation is used after preferences for mitigation strict but not in other sections)

Response authors: We have changed the text as suggested.

Table 1 first row: Change to Scale Cronbach’s Alpha

Response authors: We have changed the text as suggested.

Table 1 negative affective response row: Options presented under items (e.g., (not) anxious, etc.) are not clear – is it not anxious or is it anxious?

Response authors: We have rephrased the section in the Table for clarification.

Table 1 notes: It would be clearer if each note was on a separate line

Response authors: We have changed the text as suggested.

Lines 247-248: Change to i.e., parents(E) N = 240; parents(NE) N = 321; and individuals(NC) N = 472)

Response authors: We have changed the text as suggested.

Table 2: Should specify what the M stands for. Median?

Response authors: We have specified what M stands for in the Table title. It stands for mean. 

Table 2 notes: It would be clearer if each note was on a separate line. Should also clarify here what low, intermediate, and high mean.

Response authors: We have put each note on a separate line and clarified the operationalization of ‘ level of education’  in the Table’ s notes.

Line 250: Change to Parents(NE): parents of non-exposed…

Response authors: We have changed the text as suggested.

Line 252: Should capitalize Central Bureau of Statistics

Response authors: We have changed the text as suggested.

Lines 255-256: Reword sentence for clarity

Response authors: We have rephrased the sentence for clarity.

Table 3: It seems like it would be good to provide a bit more information in the text for the reader to know exactly what this table is showing us.

Response authors: We have added additional information in the text about Table 3. The text is now as follows: Table 3 shows the means and standard deviations of perceptions of risk (nature of the hazard,  exposure, potential health effects, probability of health effects, and negative affective response) and preferences for mitigation (regulatory mitigation preferences and strict mitigation preferences) among parents(E), parents(NE), and individuals(NC), at T1 and T2. At T1 the means of all constructs exceed the scale-median (3) to the high side of the scales, except for perceptions of the probability of health effects among individuals(NC). Notable are the considerable scores on preferences for mitigation preferences, with regard to strict measures, but especially with regard to regulatory measures. Pearson’s correlations (two-tailed) between the dependent variables vary between 0.39 and 0.75 (the correlations can be found in the Appendix).

Table 3: Footnote 2 seems like it should be in the notes of the table rather than as a footnote

Response authors: We have replaced the text from footnote 2 to the notes of Table 3.

Line 280: Change to …health effects (β=-0.29, 95% CI=-0.41/-0.17), the probability of these…

Response authors: We have changed the text as suggested.

Line 284: Remove comma after (Model 1 and 2)

Response authors: We have changed the text as suggested.

Line 291: Remove ‘show to’

Response authors: We have changed the text as suggested.

Lines 291-292: Reword sentence for clarity

Response authors: We have rephrased the sentence for clarity.

Footnotes 2 and 3: Font is not consistent. Change to None of these interaction effects were found to be significant…

Response authors: We have changed the font and the text as suggested.

Table 4: It would be clearer if each note was on a separate line

Response authors: We have put each note on a separate line.

Table 4, lines 302-303: Need a space between education and (two dummy variables). Add comma before ‘and sex’. Change to Parents(NE): parents of non-exposed children, and Individuals(NC)…

Response authors: We have changed the text as suggested.

Line 309: Use wording other than ‘change stronger’

Response authors: We have rephrased the text into ‘decreased more’.

Line 316: Extra punctuation at start of heading

Response authors: We have deleted the punctuation.

Lines 353-360: Is there some literature that can be included in this section to support the findings? Would be good to include even if it’s not specific to rubber granulate

Response authors:

The following text was added, including an additional reference supporting our findings: Literature has emphasized that affective or emotional responses to (information about) risks can differ significantly from cognitive responses to this risk (Loewenstein, Weber, Hsee, and Welch, 2001). Importantly, Loewenstein and colleagues have also emphasized that when these divergent cognitive and affective responses occur, the affective responses have likely the strongest impact on behaviour.

Line 367-368: Change to …who indicated that they were unfamiliar…

Response authors: We have changed the text as suggested.

Lines 372-373: The relationship between media and risk perceptions has been explored – and what has this shown? More references needed here and should elaborate on this a bit

Response authors: We have clarified that the study we are referring to is a literature study on the relationship between media-coverage and perceptions of risk. We have added additional information to the text. The section is now as follows: The relationship between media attention and perceptions of risk has been explored in previous studies. A literature review by Wahlberg and Sjoberg (2000) shows that current studies mainly provide evidence off a relationship between the amount of media attention about a risk topic and risk perceptions. The amount of media-attention, irrespective of its content, is shown to be associated with heightened perceived risk. This relationship between the quantity of media-attention and perceived risk can be explained by the availability-heuristic. Hence, following the availability-heuristic, the mental representations that are easily available in people’s minds are assessed as being more probable than events that are not easily mentally accessible. A high amount of media-attention on a risk topic increases the mental availability and, therefore, the perceived probability of the risk (Wahlberg and Sjoberg 2000). The peak in media attention at the start of the rubber granulate affair was probably better remembered by respondents at T1 than at T2, leading to a higher mental availability at T1 than at T2, potentially explaining the decrease of perceived risk and the preferences for mitigation between T1 and T2, irrespective of their familiarity with the results off the RIVM report.

Line 373: Reword quantity of media attention on itself can affect for clarity

Response authors: We have rephrased the text into: The amount of media-attention, irrespective of its content, is shown to be associated with heightened perceived risk.

Line 376: Check citation formatting punctuation

Response authors: We have checked the punctuation and adopted the text accordingly.

Lines 376-377: It seems like it would be better if this quote was paraphrased instead and given more context

Response authors: We have rephrased the text into: This relationship between the quantity of media-attention and perceived risk can be explained by the availability-heuristic. Hence, following the availability-heuristic, the mental representations that are easily available in people’s minds are assessed as being more probable than events that are not easily mentally accessible. A high amount of media-attention on a risk topic increases the mental availability and, therefore, the perceived probability of the risk (Wahlberg and Sjoberg 2000).

Line 384: Add ‘they’ after and

Response authors: We have changed the text as suggested.

Lines 396-398: Elaborate on the literature more to provide more information for the reader

Response authors: We have changed the text into: It seems that simply being a parent, irrespective of having a child who is exposed to rubber granulate, is related to a higher perceived risk and stronger mitigation preferences. Previous research has also suggested that parents generally perceive risks as higher than non-parents (Eibach, Libby, and Gilovich, 2003; Eibach and Mock, 2011), and more strongly support risk mitigation measures (Cameron, DeShazo, and Johnson, 2010). A possible reason for this association is that parents experience heightened levels of caution for risks in general, because they are caretakers of vulnerable beings. The study by Cameron et al. (2010) also supports this explanation as the authors found that risk aversion was higher among parents of infants than it was among parents of teenagers. An infant, after all, is more likely to be perceived as a vulnerable being than a teenager is.

Line 402: Why is this? One would think they’re serving as caregivers as well. Is this because infants are more vulnerable than teenagers? Add additional sentence after referring to Cameron 2010 study

Response authors: Our argument would be indeed because people will more likely to perceive infants as vulnerable beings than teenagers. We added the following sentence to the section: An infant, after all, is more likely to be perceived as a vulnerable being than a teenager is.

Line 405: Are there references to include about the concerns raised elsewhere?

Response authors: We have added a reference for the status report from EPA in which public concerns are mentioned, and a reference for a note with a request for clarity about the health risks of crumb rubber from the European Commission to ECHA, in which also concerns were noted.

Line 407: Don’t need e.g., - just include the refs

Response authors: We deleted ‘ e.g.’.

Lines 415-417: Reword for clarity

Response authors: We rephrased the sentence.

Line 425: Change to Our study also has a…

Response authors: We have changed the text as suggested.

Line 432: Change to …discussions of risk in a relatively short period of time

Response authors: We have changed the text as suggested.

Line 446: Remove the word ‘and’

Response authors: We have changed the text as suggested.

Line 451: Other factors such as?

Response authors: We have rephrased the sentence into: Nevertheless, other factors influencing perceptions of risk and mitigation preferences over time should not be forgotten (e.g. media attention).

Appendix: Why is this only shown for T1 and not for T2?

Response authors: We have added the correlations between variables for T2 to the Appendix.

Reviewer 2 Report

a) Style of writing needs to be improved in some areas.  I am only going to list a few. Authors need an English writer to re-read.

Line 57: change 'to' to 'that'

Line 65: change 'instigate' to 'initiate'

Line 80: needs a 'to' between 'need' and 'carefully'

Line 85: why is Risk bracketed

Line 115: Remove 'on'

Line 164: insert 'the' between 'in' and 'use'

Line 169: change "filling in" to 'completion off'

Line 183: 10.000 should be represented as 10,000

Line 116 to 118.  I would think that anxiety is sometimes affected by knowledge and belief, and so the two are not separate as implied by the sentence.

The two survey question on possible health outcomes and probability of health effects seem similar.  How are the respondents able to distinguish the difference.

Please provide additional information on the Cronbach Alfa method/results

Please clarify "negative affective response' on line 345 (should it be in brackets) as a clarifier?

Line 309: "changed stronger", please reword to were higher or are higher.

Line 395: Are you sure you mean positively influence risk perception.  Being a parent meant they had a more negative risk perception. They may be a positive relationship between the two, but that does not imply the statement above. Please check the phrasing.

Line 405-402.  Just restate the sentence as: This has led to several literature reviews......

414: Suggest what it would take to allign a person risk with the science

Author Response

Dear reviewer 2,

Thank you for your considerate comments. We have carefully considered all comments in preparing our revision. Hereby we send you our point-to-point response.

 --------------------------------------------------------------------------------------------------------------------------

Style of writing needs to be improved in some areas.  I am only going to list a few. Authors need an English writer to re-read.

Response authors: A certified corrector has checked our manuscript on correct language use and grammar and we have adapted the manuscript accordingly.

Line 57: change 'to' to 'that'

Response authors: We have changed the text as suggested.

Line 65: change 'instigate' to 'initiate'

Response authors: We have changed the text as suggested.

Line 80: needs a 'to' between 'need' and 'carefully'

Response authors: We have changed the text as suggested.

Line 85: why is Risk bracketed

Response authors: The citation applies to perceptions in general, not only perceptions of risk. We have rephrased the sentence for clarity into the following: Perceptions, including perceptions of risk, are “mental processes through which a person takes in, deals with and assesses information from the environment (physical and communicative) via the senses” (Renn, 2004, p. 1).

Line 115: Remove 'on'

Response authors: We have changed the text as suggested.

Line 164: insert 'the' between 'in' and 'use'

Response authors: We have changed the text as suggested.

Line 169: change "filling in" to 'completion off'

Response authors: We have changed the text as suggested.

Line 183: 10.000 should be represented as 10,000

Response authors: We have changed the text as suggested.

Line 116 to 118.  I would think that anxiety is sometimes affected by knowledge and belief, and so the two are not separate as implied by the sentence.

Response authors: We agree with the reviewer that cognitive and affective responses to risks interact and are therefore not entirely separate. However, literature (e.g. Risk as feelings by Loewenstein and colleagues) has shown that cognitive and affective responses to risks can differ substantially. Therefore, it is interesting to important to study both cognitive responses to risk and affective responses to risk. To clarify this section, we have rephrased it into the following: Besides from understanding the ‘cognitive’ knowledge and beliefs in the mental models of lay people, the emotional or affective response to a risk has also been emphasised in studies on perceptions of risk (Loewenstein, Weber, Hsee, and Welch, 2001; Slovic, Finucane, Peters, and MacGregor, 2004; Slovic and Peters, 2006). People do not always consider all information about a risk carefully, but often trust their feelings. It has also been shown that peoples’ feelings towards risks can differ substantially from peoples’ knowledge and beliefs about these risks. In addition, these cognitive and affective responses can have different effects on behaviour (Loewenstein et al., 2001). It is therefore important to consider not only the knowledge and beliefs of risks, but also the feelings that people experience when thinking of these risks.

The two survey question on possible health outcomes and probability of health effects seem similar.  How are the respondents able to distinguish the difference.

Response authors: We have rephrased the variable ‘possible health effects’ into ‘potential health effects’ throughout the manuscript for clarity. To answer the reviewer’s question: The survey questions included in the variable ‘potential health effects’ measure whether respondents think certain health effects could potentially follow from exposure to rubber granulate, even while these effects might only occur very sporadically. The survey questions included in the variable ‘probability of health effects’ measure the perceived probability, so the odds, of occurrence of these health effects in children exposed to rubber granulate.  

Please provide additional information on the Cronbach Alfa method/results

Response authors: We have added the following information about the Reliability Analyses: Principal Component Analyses (PCA) and Reliability Analyses were performed to develop constructs for perceptions of risk and mitigation preferences, including multiple survey questions and/or items. Constructs were considered valid if the PCA indicated that the items within a construct load on the same component, and when the internal consistency indicated by  Cronbach alpha was 0.7 or higher. Following these analyses, constructs were developed by averaging the responses to the survey questions and/or items. Table 1 shows the constructs, the corresponding survey questions, items and answer categories, and the Cronbach’s Alphas which resulted from the Reliability Analyses.

Please clarify "negative affective response' on line 345 (should it be in brackets) as a clarifier?

Response authors: We have rephrased the sentence for clarity into: The largest decreases were observed in the perceived probability of health effects caused by rubber granulate, negative affective responses to rubber granulate, and preferences for strict mitigation measures of the risk posed by rubber granulate.

Line 309: "changed stronger", please reword to were higher or are higher.

Response authors: We have rephrased the sentence into the following: Our results showed that perceived risk and preferences for mitigation decreased over time, and perceived risk and mitigation preferences decreased more among those who were familiar with new risk information following an RIVM risk assessment.

Line 395: Are you sure you mean positively influence risk perception.  Being a parent meant they had a more negative risk perception. They may be a positive relationship between the two, but that does not imply the statement above. Please check the phrasing.

Response authors: We have rephrased the sentence into the following: It seems that simply being a parent, irrespective of having a child who is exposed to rubber granulate, is related to a higher perceived risk and stronger mitigation preferences. Previous research has also suggested that parents generally perceive risks as higher than non-parents (Eibach, Libby, and Gilovich, 2003; Eibach and Mock, 2011), and more strongly support risk mitigation measures (Cameron, DeShazo, and Johnson, 2010).

Line 405-402.  Just restate the sentence as: This has led to several literature reviews......

Response authors: We have changed the text as suggested.

414: Suggest what it would take to allign a person risk with the science

Response authors: We have added the following text to the section: The application of risk communication knowledge in practice can help bridge the concerns from citizens and the conclusions following risk scientific studies (Renn and Rohrmann, 2000). A main priority should be to formulate risk communication which is adapted to the perceptions of risk in various groups to support decision making. Risk communication should address decision relevant misconceptions, complement knowledge gaps, and address the particular elements that a particular audience considers important.

Round 2

Reviewer 1 Report

Thank you for the opportunity to re-review this manuscript. Extensive changes have been made, and the manuscript has better flow and improves upon the areas that needed improvement. The authors have addressed all of the comments adequately and this is suitable for publication.

My only (very minor) comments are related to a few typos:

Line 66: remove word 'is' after risks

Line 179: need space between granulate and because

Line 185: need space between 2016 and to

Line 245: extra space between by and Cronbach's (pick Alpha or alpha and keep consistent throughout)

Line 412: sentence should read as the results of (not off)

Line 421: sentence should read as ..provide evidence of (not off)

Line 427: peoples' minds

Line 466: change off to of

Overall, e.g. and i.e. should have commas after (e.g., and i.e.,).